# Functional dichotomy and distinct nanoscale assemblies of a cell cycle-controlled bipolar zinc-finger regulator

Johann Mignolet[1*‡], Seamus Holden[2,3†], Matthieu Bergé[1†], Gaël Panis[1], Ezgi Eroglu[1], Laurence Théraulaz[1], Suliana Manley[2], Patrick H Viollier[1*]

[1]Microbiology and Molecular Medicine, Institute of Genetics and Genomics in Geneva (iGE3), Faculty of Medicine, University of Geneva, Geneva, Switzerland; [2]Laboratory of Experimental Biophysics, École Polytechnique Fédérale de Lausanne, Lausanne, Switzerland; [3]Centre for Bacterial Cell Biology, Institute for Cell and Molecular Biosciences, Newcastle University, Newcastle, United Kingdom

**\*For correspondence:** johann. mignolet@uclouvain.be (JM); patrick.viollier@unige.ch (PHV)

[†]These authors contributed equally to this work

**Present address:** [‡]Biochemistry and Molecular Genetics of Bacteria (BBGM), Université catholique de Louvain, Louvain-la-Neuve, Belgium

**Competing interests:** The authors declare that no competing interests exist.

**Abstract** Protein polarization underlies differentiation in metazoans and in bacteria. How symmetric polarization can instate functional asymmetry remains elusive. Here, we show by super-resolution photo-activated localization microscopy and edgetic mutations that the bitopic zinc-finger protein ZitP implements specialized developmental functions – pilus biogenesis and multifactorial swarming motility – while shaping distinct nanoscale (bi)polar architectures in the asymmetric model bacterium *Caulobacter crescentus*. Polar assemblage and accumulation of ZitP and its effector protein CpaM are orchestrated in time and space by conserved components of the cell cycle circuitry that coordinate polar morphogenesis with cell cycle progression, and also act on the master cell cycle regulator CtrA. Thus, this novel class of potentially widespread multifunctional polarity regulators is deeply embedded in the cell cycle circuitry.

## Introduction

Some regulatory proteins that execute important developmental, cytokinetic or morphogenetic functions are localized in monopolar fashion, whereas others are sequestered to both cell poles (*Dworkin, 2009*; *Martin and Goldstein, 2014*; *Shapiro et al., 2002*; *St Johnston and Ahringer, 2010*). It is unclear if bipolar proteins can confer specialized functions from each polar site, but examples of proteins with a bipolar disposition have been reported for eukaryotes and prokaryotes (*Davis et al., 2013*; *Martin and Berthelot-Grosjean, 2009*; *Tatebe et al., 2008*; *Treuner-Lange and Sogaard-Andersen, 2014*).

The synchronizable Gram-negative α-proteobacterium *Caulobacter crescentus* (henceforth *Caulobacter*) is a simple model system to study pole-specific organization and cell cycle control (*Tsokos and Laub, 2012*). The *Caulobacter* predivisional cell is overtly polarized and spawns two morphologically dissimilar and functionally specialized daughter cells, each manifesting characteristic polar appendages (*Figure 1A*). The swarmer progeny is a motile and non-replicative dispersal cell that samples the environment in search of food. It harbours adhesive pili and a single flagellum at one pole and is microscopically discernible from the stalked cell progeny, a sessile and replicative cell that features a stalk, a cylindrical extension of the cell envelope, on one cell pole. While the stalked cell resides in S-phase, the swarmer cell is in a quiescent G1-like state from which it only exits concomitant with the differentiation into a stalked cell. During this G1→S transition, the polar flagellum and pili of the swarmer cell are eliminated and replaced by the stalk that elaborates from the vacated cell pole. Upon sequential transcriptional activation of developmental factors during the cell

**eLife digest** Living cells become asymmetric for many different reasons and how they do so has been a long-standing question in biology. In some cells, the asymmetry arises because a given protein accumulates at one side of the cell. In particular, this process happens before some cells divide to produce two non-identical daughter cells that then go on to develop in very different ways – which is vital for the development of almost all multicellular organisms. The single-celled bacterium *Caulobacter crescentus* also undergoes this type of asymmetric division. The polarized *Caulobacter* cell produces two very different offsprings – a stationary cell and a nomadic cell that swims using a propeller-like structure, called a flagellum, and has projections called pili on its surface.

Before it divides asymmetrically, the *Caulobacter* cell must accumulate specific proteins at its extremities, or poles. Two such proteins are ZitP and CpaM, which appear to have multiple roles and are thought to interact with other factors that regulate cell division. However, little is known about how ZitP and CpaM become organized at the poles at the right time and how they interact with these regulators of cell division.

Mignolet et al. explored how ZitP becomes polarized in *Caulobacter crescentus* using a combination of approaches including biochemical and genetic analyses and very high-resolution microscopy. This revealed that ZitP accumulated via different pathways at the two poles and that it formed distinct structures at each pole. These structures were associated with different roles for ZitP. While ZitP recruited proteins, including CpaM, required for assembly of pili to one of the poles, it acted differently at the opposite pole.

By mutating regions of ZitP, Mignolet et al. went on to show that different regions of the protein carry out these roles. Further experiments demonstrated that regulators of the cell division cycle influenced how ZitP and CpaM accumulated and behaved in cells, ensuring that the proteins carry out their roles at the correct time during division. These findings provide more evidence that proteins can have different roles at distinct sites within a cell, in this case at opposite poles of a cell. Future studies will be needed to determine whether this is seen in cells other than *Caulobacter* including more complex, non-bacterial cells.

cycle (*Panis et al., 2015*), the nascent stalked cell re-establishes polarization and ultimately gives rise to an asymmetric pre-divisional cell that yield a swarmer and a stalked progeny.

The GcrA transcriptional regulator predominates in early S-phase (*Holtzendorff et al., 2004*) (*Figure 1A–B*). It accumulates during the G1→S transition and activates expression of polarity factors that are required for pilus or flagellum biogenesis and cytokinetic components (*Davis et al., 2013*; *Fioravanti et al., 2013*; *Murray et al., 2013*; *Quon et al., 1996*; *Viollier et al., 2002b*) (*Figure 1A–B*). Among GcrA target promoters, is the promoter controlling expression of the PodJ polar organizer that localizes to the pole opposite the stalk and directs assembly of the <u>C</u>aulobacter <u>p</u>ilus <u>a</u>ssembly (*cpa*) machine at that site. In this cascade, PodJ recruits the cytoplasmic CpaE protein that then promotes the localization and assembly of CpaC secretin localization (*Figure 1B*) (*Viollier, 2002a*). Another key promoter controlled by GcrA is the one driving expression of the master cell cycle regulator CtrA that induces the synthesis of a second wave of polar and morphogenesis factors in late S-phase including the *cpa* operon (*Figure 1B*). The abundance of CtrA and GcrA is regulated at the level of synthesis and degradation (*Collier et al., 2006*; *Domian et al., 1997*) and as a result, cell division spawns a swarmer and stalked cell progeny containing CtrA and GcrA, respectively.

An important polarity determinant in the α-proteobacteria is the conserved matrix protein PopZ (*Figure 1A*) that organizes poles by forming a molecular lattice that traps polar determinants and effectors (*Bowman et al., 2008*; *Deghelt et al., 2014*; *Ebersbach et al., 2008*; *Grangeon et al., 2015*; *Laloux and Jacobs-Wagner, 2013*). PopZ is bipolar in the *Caulobacter* predivisional cell and it interacts directly with numerous cell cycle kinases, the ParAB chromosome segregation proteins and cell fate determinants (*Holmes et al., 2016*). Here, we dissect at the genetic and cytological level the polar localization and function of two poorly characterized trans-membrane proteins, the zinc-finger protein ZitP and the CpaM effector protein, that are polarly localized and that execute

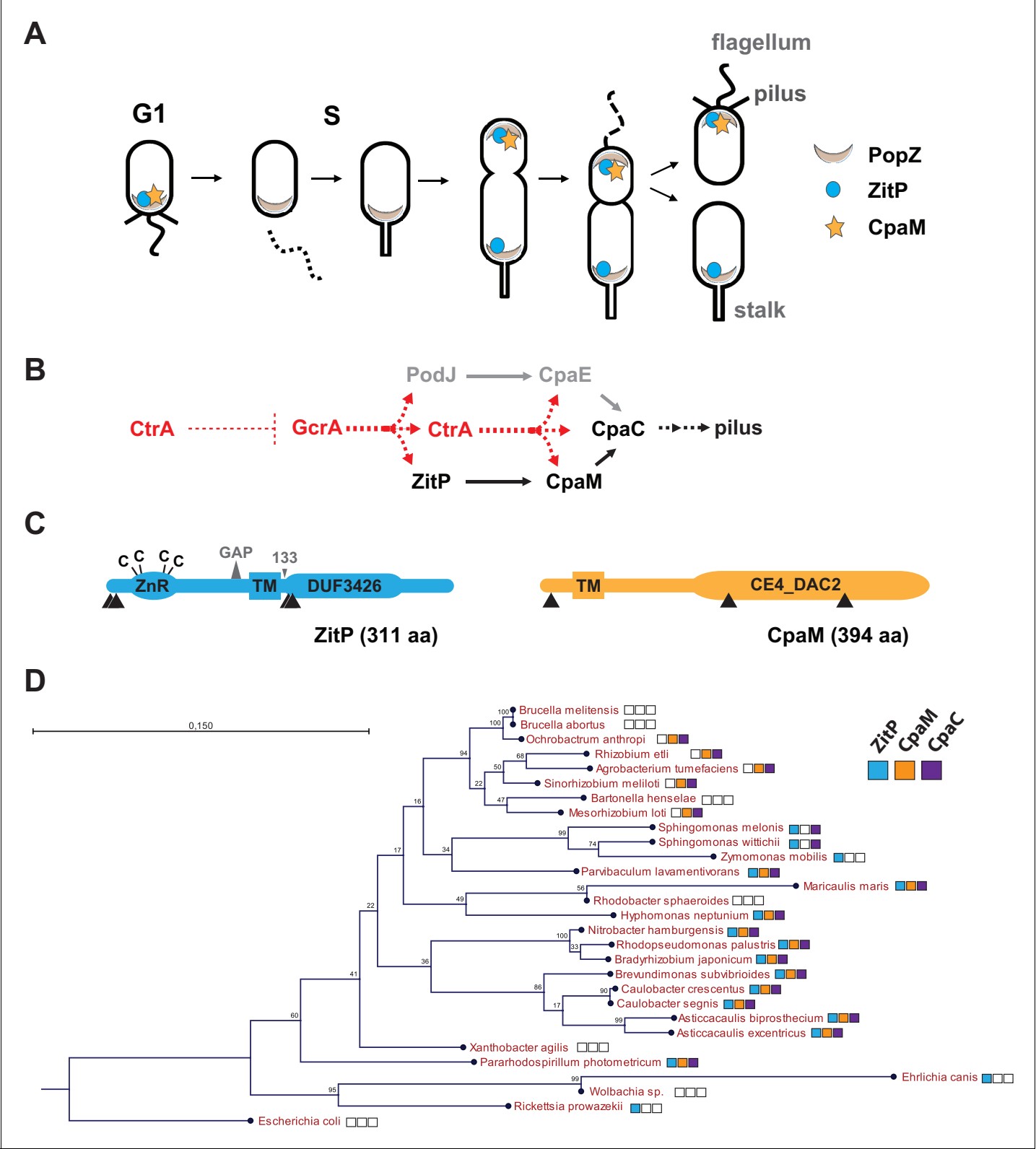

**Figure 1.** Cell cycle profile and phylogeny of ZitP and CpaM. (**A**) Scheme depicting the polarized factors PopZ, ZitP and CpaM during the cell cycle of the dimorphic bacterium *C. crescentus*. (**B**) Pilus assembly pathways and global dependencies of the two master cell cycle regulators GcrA and CtrA on the expression of the polar factors PodJ, CpaE, ZitP, CpaM and CpaC that control pilus biogenesis. Red and black dashed lines highlight transcriptional activation and polar recruitment, respectively. (**C**) Schematic representation (drawn to scale) of ZitP (blue) and CpaM (yellow). ZnR: zinc finger domain;
*Figure 1 continued on next page*

*Figure 1 continued*

TM: transmembrane domain, C: cysteine. Arrowheads below each protein pinpoint the site of truncation due to transposon insertion in the coding sequence. The large triangle on top of ZitP shows the 2 amino acid residues deleted in the ZitP$^{GAP}$ variant and the small triangle depicts the position of residue 133 where the ZitP coding sequence is truncated in the ZitP$^{1-133}$ variant. (D) Conservation of ZitP (blue), CpaM (yellow) and CpaC (purple) across the α-proteobacterial clades. The phylogenetic tree was built in CLC Main Workbench (http://www.clcbio.com/products/clc-main-workbench/) from 16S RNA alignments based on the Neighbor Joining method (Juke Cantor substitution model) with 100 bootstrap replicates. Empty boxes mean that no ortholog was found in the genome. Scale bar, 0.15 substitution per site.

multiple regulatory functions. We unearthed two separate localization pathways for each cell pole, one PopZ-dependent and another that is PopZ-independent, and we provide evidence by photo-activated localization microscopy (PALM) and by genetic dissection that each polar cluster has a distinctive architecture and a specialized function.

## Results

### ZitP and CpaM are required for pilus biogenesis.

As pili are necessary for infection by the lytic caulophage CbK (φCbK) (*Skerker and Shapiro, 2000*), we specifically sought mutants in pilus assembly factors encoded outside of the major pilus assembly *cpa* gene locus (*pilA-cpaA-K*) (*Christen et al., 2016*; *Skerker and Shapiro, 2000*). To this end, we conducted *himar1*-transposon (*Tn*) mutagenesis of wild-type (*WT*) *Caulobacter* in the presence of φCbK (see Methods) and recovered mutants with Tninsertions in *CCNA_02298*, renamed here *zitP* (*z*inc-finger targeting the poles) because of the pleiotropic roles detailed below, or in *cpaM* (*CCNA_03552*) (*Figure 1C*) (*Marks et al., 2010*). While both genes have previously been implicated in polar functions and their transcription is cell cycle-regulated (*Christen et al., 2016*; *Fioravanti et al., 2013*; *Fumeaux et al., 2014*; *Hughes et al., 2010*; *McGrath et al., 2007*), they are poorly characterized. The *zitP* gene is predicted to encode a 311-residue bitopic trans-membrane (TM) protein harbouring a CXXC-(X)$_{19}$-CXXC motif that binds a zinc ion (zinc_ribbon_5 or PF13719 superfamily, residues 1-37) at the cytoplasmic N-terminus (*Bergé et al., 2016*) and a conserved domain-of-unknown function (DUF3426, residues 128-245) in the C-terminal region that is predicted to reside in the periplasm (*Figure 1C*). The *cpaM* gene codes for a 394-residue protein harbouring a single N-terminal TM domain and a C-terminal CE4_DAC2-like polysaccharide deacetylase domain predicted to be periplasmic (*Figure 1C*). ZitP and CpaM are not restricted to the *Caulobacter* lineage as BLASTP searches revealed orthologs in many α-proteobacterial clades (*Figure 1D*). To confirm the phenoytpes of the *Tn* insertion mutants, we engineered strains with an in-frame deletion in *zitP* (Δ*zitP*) or *cpaM* (Δ*cpaM*) and found that the mutants no longer supported plaque formation (lysis) by the pilus-specific bacteriophage φCbK. By contrast, plaques were still formed by the S-layer specific caulophage φCr30 (*Edwards and Smit, 1991*) (*Figure 2A*), showing that mutations in *cpaM* or *zitP* prevent infection of φCbK, but not all phages. This defect was corrected upon expression of either ZitP or CpaM from an ectopic locus in Δ*zitP* or Δ*cpaM* cells, respectively (*Figure 2A*).

Next, we conducted time-course adsorption assays and found the adsorption kinetics of Δ*zitP* and Δ*cpaM* cells to be substantially compromised compared to *WT* cells (*Figure 2B*). The φCbK adsorption kinetics of the mutants closely resemble those for Δ*cpaC* cells that cannot assemble pili because they lack the CpaC secretin (*Skerker and Shapiro, 2000*). Moreover, immunoblotting revealed that Δ*zitP* and Δ*cpaM* cells do not accumulate the modified form of CpaC, CpaC* (*Figure 2C–D*). A comparable reduction in CpaC* abundance has been previously reported for Δ*cpaE*, Δ*podJ* and Δ*pleA* cells that no longer assemble a polar CpaC pilus channel in the outer membrane and cannot be infected by φCbK (*Viollier and Shapiro, 2003*; *Viollier et al., 2002b*). However, CpaC* accumulates in Δ*pilA* cells (*Figure 2C*), suggesting that the CpaC channel forms independently of PilA. To test whether Δ*zitP* and Δ*cpaM* cells assemble a pilus filament on the cell surface, we conducted shearing assays followed by immunoblotting using antibodies to the PilA pilin, the subunit of the pilus filament (*Figure 2E*) (*Skerker and Shapiro, 2000*). Whereas PilA was efficiently released from *WT* cells into the supernatant by shearing, no PilA was detectable in the

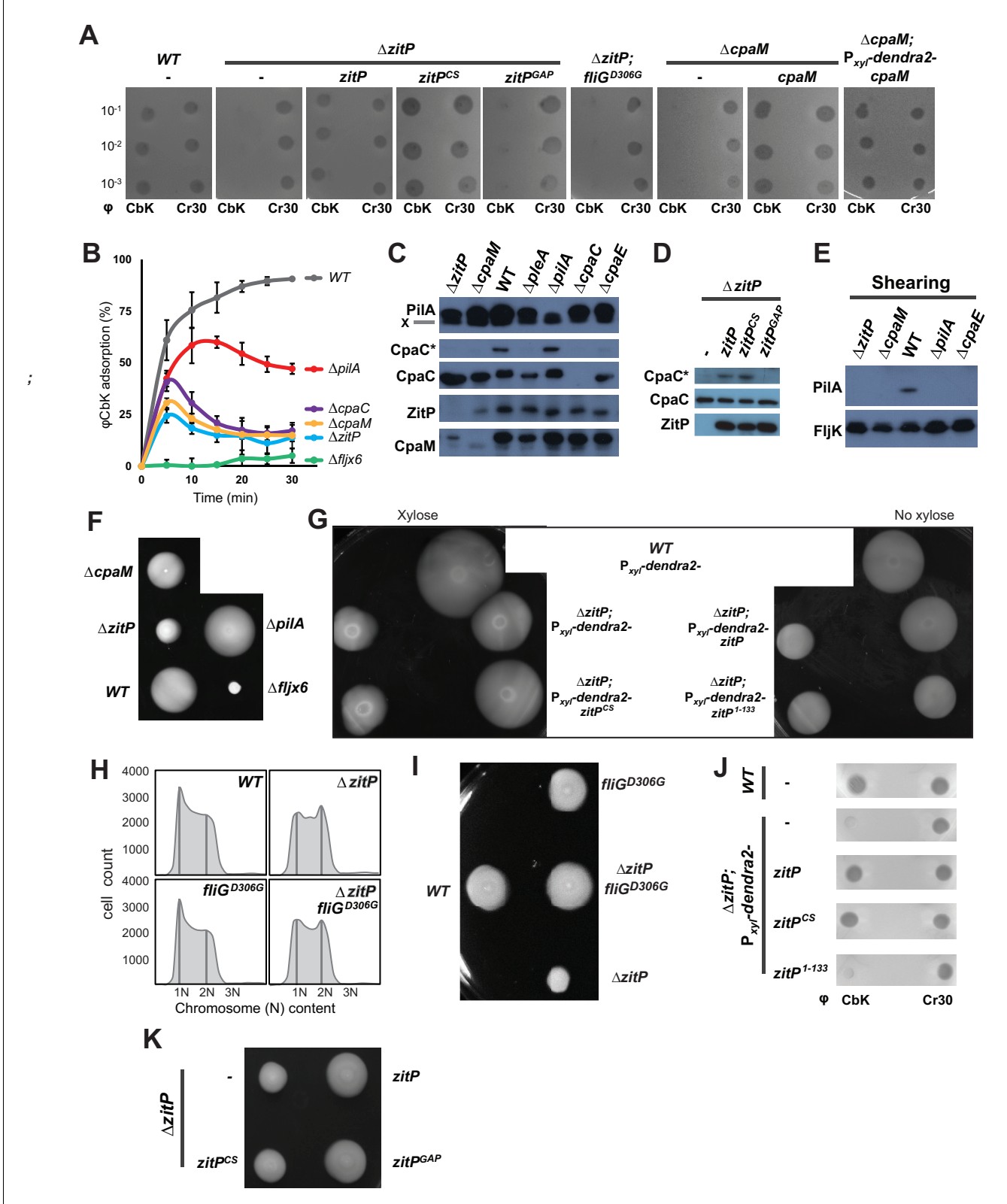

**Figure 2.** Functional dichotomy in ZitP and effects on polar morphogenesis. (**A**) Bacteriophage infection assays of *WT*, Δ*zitP*, Δ*zitP;fliG*[D306G] and Δ*cpaM* mutant cells. Cells harbour empty pMT335 or a complementing plasmid (pMT335 backbone) and were grown in the absence of vanillate. No xylose was added to the agar for the phage assay on Δ*cpaM*; P$_{xyl}$-*dendra2-cpaM* cells. The phages φCbK and φCr30 were spotted with serial dilution on *C. crescentus* embedded in top agar. Sensitivity to phages is indicated by plaques (lysis). (**B**) Adsorption kinetics of φCbK to *WT* and mutant cells. (**C**)
*Figure 2 continued on next page*

*Figure 2 continued*

Steady-state levels of ZitP, CpaM, CpaC, modified CpaC (CpaC*) and PilA in *WT* and mutant cells as determined by immunoblotting. In the PilA immunoblots, the asterisk (*) points to a non-specific band. (D) Immunoblots showing the steady-state levels of monomeric CpaC and CpaC* in *ΔzitP* cells harbouring pMT335 or derivatives encoding ZitP$^{WT}$, ZitP$^{CS}$ or ZitP$^{GAP}$ grown in the presence of vanillate (50 μM). (E) Immunoblots showing PilA and FljK abundance in supernatants of *WT* and various mutant cells. Supernatants were harvested from mid-log cultures after shearing. (F) Swarming motility test performed on soft (0.3%) agar with *WT*, *ΔzitP*, *ΔcpaM*, *ΔpilA* and *Δfljx6* mutant cells. (G) Complementation of the motility defect on swarm (0.3%) agar displayed by the *ΔzitP* cells expressing Dendra2-ZitP variants from P$_{xyl}$ at the *xylX* locus. Xylose was added to the swarm (0.3%) agar as indicated. (H) Flow cytometry of exponential phase *WT* and *ΔzitP* cells. N refers to chromosome equivalents. (I) Suppression of the *ΔzitP* motility phenotype by *fliG*$^{D306G}$ point mutation as shown on a swarm (0.3%) agar plate. (J) Phage spot tests with φCr30 and φCbK on *WT* or *ΔzitP* cells expressing Dendra2-ZitP variants from P$_{xyl}$ at the *xylX* locus. Cells were embedded in top agar containing xylose (0.3%). (K) Motility assays of *ΔzitP* cells expressing WT ZitP (ZitP$^{WT}$), ZitP$^{CS}$ or ZitP$^{GAP}$ from pMT335. Swarming motility was assessed in absence of vanillate on 0.3% agar.
The following figure supplements are available for figure 2:

**Figure supplement 1.** Master regulator-dependent promoters in *ΔzitP*.
**Figure supplement 2.** CtrA- and (p)ppGpp-independent influence of the *ΔzitP* motility defect.

supernatants of *ΔcpaE*, *ΔzitP* and *ΔcpaM* cells after shearing (*Figure 2E*), even though PilA is clearly expressed in these cells (*Figure 2C*). As the major subunit of the flagellar filament, the FljK flagellin, accumulates in the supernatants in all samples (*Figure 2E*), we conclude that ZitP and CpaM are required for the presentation of PilA on the cell surface and, as shown below, that they act in the same pathway (*Figure 1B*).

## Control of motility, G1-phase and the CtrA regulon.

The φCbK adsorption kinetics hinted that motility might be altered in *ΔzitP* and *ΔcpaM* cells. This hypothesis is based on the comparison of the φCbK adsorption kinetics to *WT*, *ΔpilA* and *Δfljx6* (lacking all six flagellin genes: *fljJ/K/L/M/N/O*) cells to *ΔzitP* and *ΔcpaM* cells. While pililess *ΔpilA* cells assemble a flagellum and are motile (*Figure 2F*), *Δfljx6* cells are flagellumless, but piliated (φCbK sensitive) (*Guerrero-Ferreira et al., 2011*). The kinetics of adsorption of φCbK to *ΔzitP* and *ΔcpaM* cells was strongly reduced compared to *WT*, fitting halfway between the adsorption curves of φCbK to *ΔpilA* and *Δfljx6* cells (*Figure 2B*). Since it is known that φCbK first reversibly adsorbs to the flagellar filament rotating counter-clockwise, before the irreversibly attachment to the pilus portal is established for productive infection (*Guerrero-Ferreira et al., 2011*), we wondered whether there are fewer motile cells in the *ΔzitP* and *ΔcpaM* populations than in *WT* or if motility in these mutants is altered in other ways. In fact, motility tests on swarm (0.3%) agar revealed a mild reduction in motility of *ΔcpaM* cells and a severe reduction of *ΔzitP* cells compared to *WT* (*Figure 2F*). However, *ΔzitP* cells still have residual motility that allows them to spread in swarm agar compared to *Δfljx6* cells (*Figure 2F*). Expression of Dendra2-ZitP from an ectopic locus confers near WT motility to *ΔzitP* cells (*Figure 2G*), showing that this deficiency in motility is indeed due to the absence of ZitP.

As *Caulobacter* divides into a motile G1-phase cell and a sessile S-phase cell, mutants accumulating fewer G1-phase cells in the population can exhibit reduced motility on soft agar (*Sanselicio et al., 2015*; *Sanselicio and Viollier, 2015*). To test if ZitP controls the G1 cell number, we used flow cytometry to quantify the number of G1 cells and indeed observed fewer G1 cells in the *ΔzitP* population compared to *WT* (*Figure 2H*). Knowing that the master cell cycle transcriptional regulator CtrA retains cells in G1-phase and activates many cell cycle-regulated promoters that fire in G1-phase (*Domian et al., 1997*; *Fumeaux et al., 2014*; *Quon et al., 1996*), we then conducted promoter-probe assays using several CtrA-activated promoters fused to the promoterless *lacZ* gene and quantified CtrA-dependent promoter activity in *WT* and *ΔzitP* cells (*Figure 2—figure supplement 1*). While all such promoter-probe reporters for the CtrA regulon exhibited a decrease in activity by 30-40% in *ΔzitP* versus *WT* cells, promoter-probe reporters for the GcrA regulon or other reporters were unaffected. Thus, ZitP is required for optimal CtrA activity and G1 cell accumulation.

The reduction in CtrA-dependent transcription does not appear to be solely responsible for the motility defect of *ΔzitP* cells. First, promoter-probe assays revealed that *ΔcpaM* cells also suffer from reduced CtrA-dependent activation (*Figure 2—figure supplement 2A*), even though their motility

exceeds that of Δ*zitP* cells (*Figure 2F*). Second, we were able to mitigate the defect in CtrA-dependent transcription by ectopic expression of the (p)ppGpp alarmone, a signalling molecule that enhances CtrA function and stability via a poorly understood mechanism (*Gonzalez and Collier, 2014*). We accomplished this by heterologously expressing the truncated version of the *E. coli* (p)ppGpp-synthase RelA (RelA') from the xylose-inducible promoter at the *xylX* locus in *WT* and Δ*zitP* cells. LacZ-based promoter-probe assays revealed that ectopic induction of (p)ppGpp restores CtrA-dependent promoter activity to near WT levels (*Figure 2—figure supplement 2B*). However, the motility of Δ*zitP* cells ectopically producing (p)ppGpp is still substantially lower than that of *WT* cells (*Figure 2—figure supplement 2C–D*), indicating that ZitP also promotes motility through a CtrA- and (p)ppGpp-independent pathway.

To reinforce this conclusion, we isolated a spontaneous motile suppressor of Δ*zitP* cells (see Materials and Methods, *Figure 2I*) with a single point mutation in the *fliG* flagellar gene (*fliG^{D306G}*) that neither corrects the pilus assembly defect (φCbK-resistance, *Figure 2A*), nor the reduction in G1 cell number of the Δ*zitP* mutant (*Figure 2H*). As FliG encodes a component of the flagellar motor that is associated with the cytoplasmic membrane (*Macnab, 2003*), we conclude that ZitP controls pilus biogenesis and a multifactorial motility phenotype, with a minor contribution from a CtrA-dependent pathway and a major one from a CtrA-independent pathway(s) that can be bypassed by a mutant variant of FliG.

## Distinct polar ZitP assemblies control CpaM localization

To investigate if ZitP also controls its polar functions from the cell pole, we resorted to live-cell fluorescence imaging by epifluorescence microscopy (*Figure 3—figure supplement 1A–D*) and photo-activated localization microscopy (PALM, *Figure 3A–B and D–E*) (*Betzig et al., 2006*) using *WT*, Δ*zitP* or Δ*cpaM* cells expressing functional Dendra2-CpaM or Dendra2-ZitP. We observed Dendra2-ZitP to adopt a bipolar disposition in dividing cells, whereas Dendra2-CpaM is restricted to the pole opposite the stalk where the pilus biogenesis machinery assembles (*Figure 3A–B*; *Figure 3—figure supplement 2A–C*). While Dendra2-ZitP localization is not noticeably perturbed in Δ*cpaM* cells (*Figure 3—figure supplement 1B–C*), Dendra2-CpaM is dispersed in Δ*zitP* cells (*Figure 3A*; *Figure 3—figure supplement 1D and 2B*). Moreover, biochemical pull-down experiments with ZitP-TAP (*Figure 3—figure supplement 3*) and reciprocal co-immunoprecipitation experiments using antibodies to ZitP and CpaM (*Figure 3C*) showed that ZitP and CpaM reside in a complex. Since Dendra2-ZitP and Dendra2-CpaM localization is not affected in Δ*podJ*, Δ*cpaE* or Δ*cpaC* cells (*Figure 3F*, *Figure 3—figure supplement 1C and D*) and since CpaE localization is not noticeably altered in Δ*zitP* and Δ*cpaM* cells (*Figure 3—figure supplement 4A–B*), we conclude that ZitP and CpaM are part of a previously unknown (PodJ/CpaE-independent) polarization pathway for pilus assembly in *Caulobacter* in which ZitP recruits CpaM (*Figure 1B*).

PALM analysis disclosed differently shaped and sized complexes of Dendra2-ZitP at each *Caulobacter* pole. Both Dendra2-ZitP clusters appear extended, suggesting that ZitP multimerization along the polar membrane is spatially restricted (*Figure 3A–B*; *Figure 3—figure supplement 2A and C*). Quantification of the 2D area and shape-based analyses (circularity, solidity and eccentricity) showed that ZitP clusters extending into the base of the stalk are significantly larger and differently shaped than the extended fluorescent foci lining the cap of the opposite (swarmer) pole (*Figure 3B and D*; *Figure 3—figure supplement 2A, C* and *5A–D*). In further support of the existence of two distinct nanostructures of ZitP at each pole, genetic experiments revealed that different pathways govern ZitP polarization: one requiring PopZ and another operating independently of PopZ. Imaging of Dendra2-ZitP in Δ*popZ* cells revealed mainly monopolar foci (*Figure 3B and F*; *Figure 3—figure supplement 1C* and *2C*), resembling those seen at the pole opposite the stalk in *WT* cells (*Figure 3B and D*; *Figure 3—figure supplement 2C* and *5C–D*). Quantitative analysis of the polar residence time using stroboscopic single particle tracking PALM (*Gebhardt et al., 2013*) revealed a strong reduction in polar binding times of Dendra2-ZitP in Δ*popZ* compared to that of *WT* cells (*Figure 3E*; *Figure 3—figure supplement 6A–D*). Thus, PopZ promotes the formation of a large polar ZitP assembly at the stalked pole, whereas a small complex of ZitP sequesters independently of PopZ at the opposite pole.

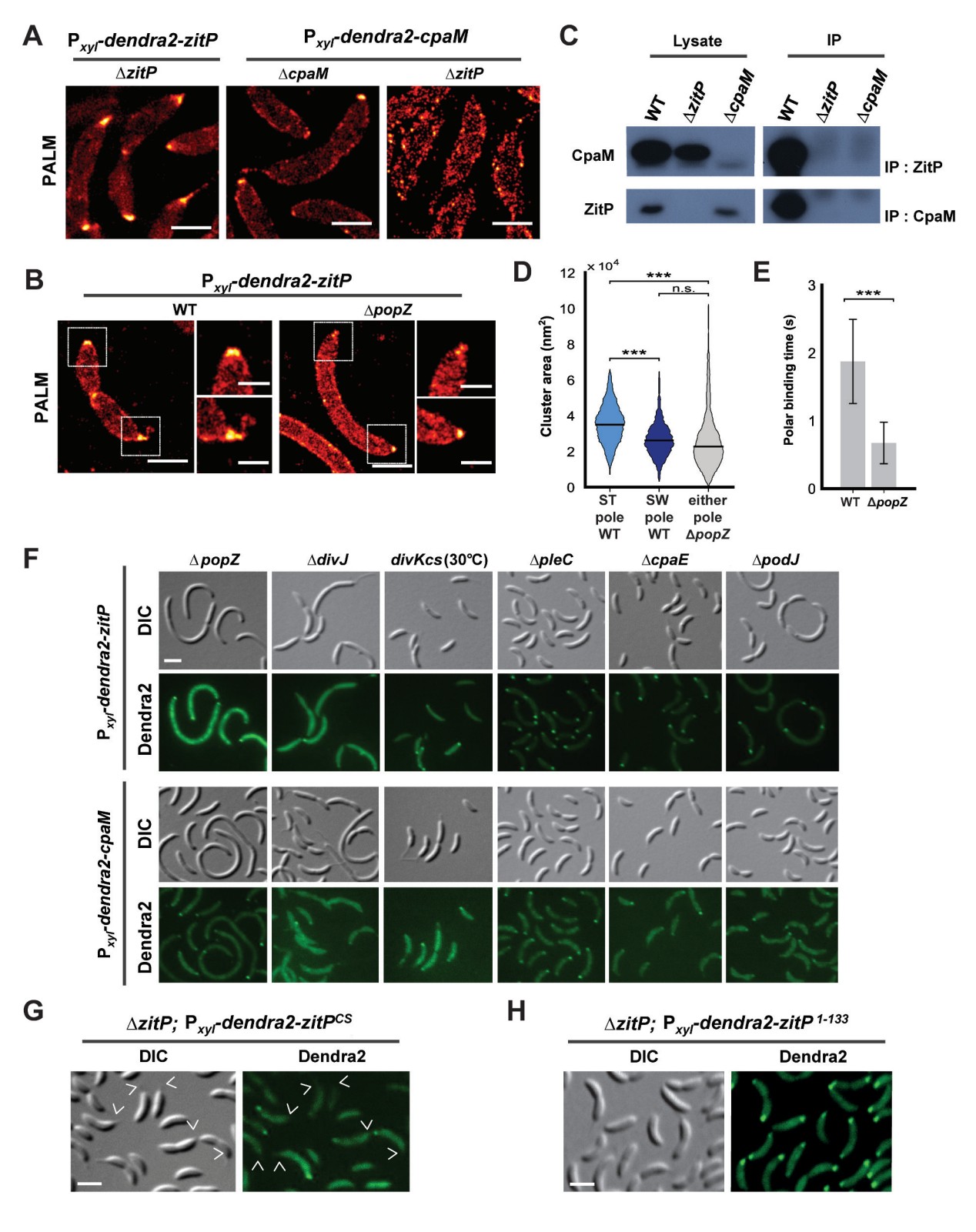

**Figure 3.** Distinct ZitP nanoscale assemblies and localization determinants. (A) Photo-activated light microscopy (PALM) imaging of Dendra2-ZitP or Dendra2-CpaM expressed from the xylose-inducible P$_{xyl}$ promoter on a plasmid integrated at the chromosomal *xylX* locus in Δ*zitP* or Δ*cpaM* cells exposed to xylose 3 hours before imaging. Scale bar: 1 μm. (B) PALM imaging of Dendra2-ZitP in *WT* or Δ*popZ*::Ω cells. We induced expression of Dendra2-ZitP from the xylose-inducible P$_{xyl}$ promoter on a plasmid integrated at the chromosomal *xylX* locus by the addition of xylose 3 hours before
*Figure 3 continued on next page*

*Figure 3 continued*

imaging. Scale bar: 1 μm. Scale bar of zoomed images: 0.5 μm. (C) Co-immunoprecipitation (co-IP) of ZitP or CpaM with polyclonal antibodies to CpaM or ZitP, respectively. Immunoprecipitates and cell lysates from *WT*, Δ*zitP* or Δ*cpaM* cells were probed for the presence of ZitP or CpaM. (D) Projected area of the Dendra2-ZitP polar complex as determined by PALM from Dendra2-ZitP expressed in *WT* and Δ*popZ*::Ω cells. Black lines indicate medians. Statistical significance from Mood's median test: *n.s,* p>*0.05;* ***p<0.001. (E) ZitP polar binding times in *WT* and Δ*popZ*::Ω cells, measured via single particle tracking PALM. Error bars indicate 95% confidence interval of the fit to the data (*Figure 3—figure supplement 6D*). Statistical significance from a 2 sample t-test: ***p=p<0.001. (F) Epifluorescence (Dendra2) and Nomarski (DIC) images depicting the localization of Dendra2-ZitP or Dendra2-CpaM in Δ*popZ*::Ω, Δ*divJ, divKcs, ΔpleC, ΔcpaE* or Δ*podJ* cells. Expression of Dendra2-ZitP or Dendra2-CpaM was induced from the chromosomal *xylX* locus with xylose 4 hours before imaging. Scale bars: 1 μm. (G) (H) Epifluorescence (Dendra2) and Nomarski (DIC) images depicting the localization of the motility-deficient and pilus-proficient Dendra2-ZitP$^{CS}$ variant (G) or the motility-proficient and pilus-deficient Dendra2-ZitP$^{1-133}$ variant (H) in Δ*zitP* cells. Arrow heads pinpoint stalked poles. We induced expression of Dendra2-fusions from the xylose-inducible P$_{xyl}$ promoter on a plasmid integrated at the chromosomal *xylX* locus by the addition of xylose 4 hours before imaging. Scale bars: 1 μm.

The following figure supplements are available for figure 3:

**Figure supplement 1.** Extrinsic determinant for the localization of ZitP and CpaM.

**Figure supplement 2.** ZitP and CpaM polar localization by PALM.

**Figure supplement 3.** Tandem affinity purification of ZitP.

**Figure supplement 4.** CpaE localization in Δ*zitP* and Δ*cpaM* mutant cells.

**Figure supplement 5.** Quantitative analysis of ZitP cluster shape and area.

**Figure supplement 6.** Binding time estimation by stroboscopic single particle tracking of ZitP.

**Figure supplement 7.** Intrinsic determinants for ZitP localization and function.

**Figure supplement 8.** Effect of DUF3426 on ZitP function.

## Localization and functional determinants in ZitP

To identify the determinants within ZitP governing the differential polar localization and to test if they support specific functions, we first constructed a mutant variant of ZitP in which all four zinc-coordinating cysteine residues in the zinc-finger domain (*Bergé et al., 2016*) are replaced by serine residues (henceforth ZitP$^{CS}$, *Figure 1C*). The motility of Δ*zitP* cells expressing ZitP$^{CS}$ or Dendra2-ZitP$^{CS}$ is reduced compared to those expressing the WT version of ZitP (ZitP or Dendra2-ZitP; *Figure 2G and K*). While Dendra2-ZitP$^{CS}$ exclusively localizes to the pole opposite the stalk in Δ*zitP* cells (*Figure 3G*; *Figure 3—figure supplement 7A*), it still supports lysis by φCbK (*Figure 2A*) and CpaC* assembly (*Figure 2D*). ZitP$^{CS}$ supports localization of Dendra2-CpaM to the pole opposite the stalk and co-immunoprecipitation experiments show that it interacts with CpaM (*Figure 3—figure supplement 7B–D*). ZitP$^{CS}$ also confers (CpaM-dependent) firing of CtrA-activated promoters with similar efficiency as WT ZitP (*Figure 3—figure supplement 8A*). Since Dendra2-CpaM is also still monopolar in Δ*popZ* cells, zinc-binding within the zinc_ribbon_5 domain is necessary for the interaction between PopZ and ZitP (*Bergé et al., 2016*), but not for CpaM localization/interaction. Thus, inactivation of the zinc-coordinating residues in ZitP effectively mimics the monopolar localization of Dendra2-ZitP in Δ*popZ* cells and functions as unmodified ZitP with respect to the functions that depend on CpaM.

By contrast, the opposite effect was seen when ZitP$^{1-133}$, a ZitP variant that lacks the periplasmic DUF3426 but retains the cytoplasmic and TM domains (residues 1-133, *Figure 1C*), is expressed in Δ*zitP* cells. ZitP$^{1-133}$ supports efficient motility and is polarly localized, but no longer supports pilus function (i.e. plaque formation by φCbK), CpaM localization and efficient CtrA-activated transcription (*Figure 2G and J*, *Figure 3—figure supplement 8A–B*). Thus, the periplasmic DUF3426 plays a critical role in promoting pilus assembly through the polar recruitment of CpaM.

Support for the notion that DUF3426 function is regulated from sequences N-terminal to the DUF3426 came from a forward genetic screen (see Materials and Methods) that led to the

identification of ZitP$^{GAP}$ (*Figure 1C*), a mutant variant in which residues Arg93 and Ala94 preceding the TM domain are deleted. ZitP$^{GAP}$ supports motility (*Figure 2K*), but neither plaque formation by φCbK, nor CpaC* production (*Figure 2A and D*). As ZitP$^{GAP}$ still localizes to the cell poles, interacts with CpaM and recruits Dendra2-CpaM (*Figure 3—figure supplements 7A–D* and *8C*), ZitP also acts on pilus biogenesis independently of CpaM localization.

Taken together our experiments indicate that function and localization of ZitP can be genetically uncoupled. The periplasmic DUF3426 region is required for pilus biogenesis and CtrA-dependent transcription and it implements these functions via the recruitment of CpaM to the pole opposite the stalk. The zinc_ribbon_5 domain promotes PopZ-dependent localization of ZitP to the stalked pole and efficient swarming motility by an unknown mechanism. Interestingly, in a related study, we recently found that ZitP controls PopZ localization independently of the DUF3426 (*Bergé et al., 2016*).

## Cell cycle control of ZitP and CpaM assemblies

Synchronization studies and genetic experiments with cell cycle mutants showed that ZitP and CpaM polarization is temporally and functionally coordinated with cell cycle progression. Immunoblotting revealed the steady-state levels of ZitP and CpaM to fluctuate during the cell cycle (*Figure 4A*), exhibiting a trough during the G1→S transition and concomitant loss of polar fluorescence at this time (*Figure 4B–C*). Consistent with the genetic and cytological hierarchy, ChIP-Seq data shows that the early S-phase regulator GcrA directly promotes ZitP and CtrA expression, while the late S-phase regulator CtrA activates expression of CpaM (*Fiebig et al., 2014*; *Fioravanti et al., 2013*; *Fumeaux et al., 2014*; *Murray et al., 2013*). Moreover, ZitP, CtrA and CpaM abundance is reduced when GcrA is depleted or inactivated (*Figure 4D*). ZitP expression is also strongly reduced in the absence of the CcrM adenine methyltransferase that methylates adenines at the N6-position in the context of 5'-GANTC-3' sequences. GANTC methylation is required for efficient recruitment of GcrA to its target promoters (*Fioravanti et al., 2013*; *Murray et al., 2013*).

Additionally, we found that the DivJ-PleC-DivK (kinase-phosphatase-substrate) system that regulates cell cycle progression and polar development influences the appearance of polar Dendra2-ZitP and Dendra2-CpaM (*Figure 3F*, *Figure 3—figure supplement 1C–D*). Specifically examining the localization in mutants where the phosphoflux is shifted towards the accumulation of the phosphorylated form of the DivK cell fate determinant (*Tsokos and Laub, 2012*), we found that such a mutation (inactivation of the PleC phosphatase, ΔpleC) promotes ZitP/CpaM polarization as indicated by the bipolar localization of Dendra2-CpaM. By contrast, mutations that have the opposite effect on DivK activity or DivK phosphorylation (caused by the *divK$^{CS}$* or Δ*divJ* mutation), disfavour Dendra2-ZitP (but not Dendra2-CpaM) polarization (*Figure 3F*, *Figure 3—figure supplement 1C–D*). Thus, polar reprogramming of ZitP and CpaM is deeply integrated into the *Caulobacter* cell cycle through conserved components of the α-proteobacterial cell cycle (*Brilli et al., 2010*).

## Discussion

The pole-specific and distinctly shaped assemblies of ZitP are governed via independent localization pathways and linked with functional specialization (*Figure 4E*). While ZitP acts on pilus assembly by recruiting CpaM and, subsequently, the CpaC pilus channel to the pole opposite the stalk (*1B* and *4E*), CpaM is also required for efficient activation of CtrA-dependent promoters by an unknown mechanism. A similar reduction in CtrA-dependent transcription occurs in Δ*zitP* cells that are unable to localize CpaM. While diminished CtrA activity can undermine motility by reducing the number of motile G1-phase cells in the population (*Sanselicio et al., 2015*; *Sanselicio and Viollier, 2015*), ZitP affects motility in another way, since Δ*zitP* cells are diminished in motility compared to Δ*cpaM* cells. Moreover, ectopic induction of the alarmone (p)ppGpp reinforces CtrA abundance and activity (*Boutte et al., 2012*; *Gonzalez and Collier, 2014*; *Lesley and Shapiro, 2008*; *Ronneau et al., 2016*; *Sanselicio and Viollier, 2015*), but only modestly improves the motility of Δ*zitP* cells.

Such a motility defect also manifests when ZitP$^{CS}$, a variant that no longer localizes to the stalked pole, is expressed in Δ*zitP* cells. How ZitP promotes swarming motility from the stalked pole is unclear, but there is precedence of other regulators (SpmX/Y and CpdR) that localize exclusively to the stalked pole and affect *Caulobacter* motility indirectly by regulating cell cycle factors (*Janakiraman et al., 2016*; *McGrath et al., 2006*; *Radhakrishnan et al., 2008*). Moreover, SpmX

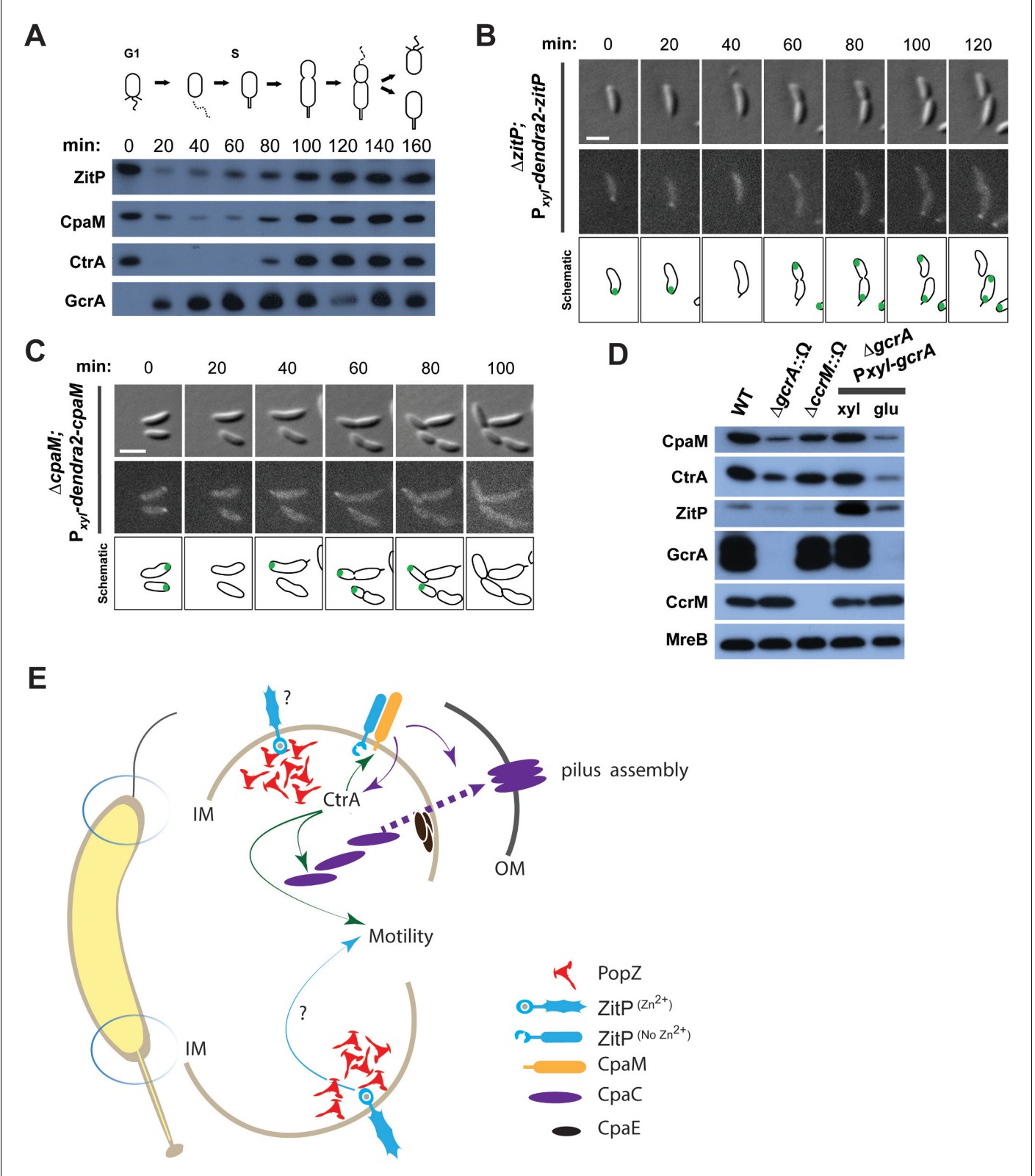

**Figure 4.** Cell cycle regulation of ZitP and CpaM localization. (**A**) Immunoblots showing the levels of ZitP, CpaM and master cell cycle regulators along the *C. crescentus* cell cycle in a synchronized *WT* population. The upper scheme depicts *C. crescentus* cell cycle stages. (**B**) (**C**) Epifluorescence (Dendra2) and Nomarski (DIC) images depicting the localization of Dendra2-ZitP (**B**) and Dendra2-CpaM (**C**) in synchronized Δ*zitP* or Δ*cpaM* cells, respectively. We induced expression of Dendra2 fusions expressed from the xylose-inducible P*xyl* promoter on a plasmid integrated at the

*Figure 4 continued on next page*

*Figure 4 continued*

chromosomal *xylX* locus. Schematic drawings highlight Dendra2 localizations. After synchronization, cells were resuspended in M2G and imaged every 20 minutes. Scale bars: 1 µm. (D) Steady-state levels of ZitP, CpaM, CtrA, GcrA, CcrM and MreB (control) in *WT*, *gcrA* and *ccrM* mutant cells. Xylose (0.3%, xyl) or glucose (0.2%, glu) were supplemented to the medium in order to induce/deplete GcrA in Δ*gcrA xylX*::P$_{xyl}$-*gcrA* cells. (E) Schematic representation of the two *Caulobacter* cell poles. At the stalked pole, the PopZ matrix promotes the recruitment of ZitP. The Zn$^{2+}$-bound zinc-finger domain of ZitP prevents ZitP/CpaM association and influences CtrA activity and swarming motility. At the opposite pole, the inactive Zn$^{2+}$-unbound zinc-finger domain allows the formation of the ZitP/CpaM complex and the export and assemblage of CpaC in the outer membrane (OM) independently of PopZ.

and CpdR interact with PopZ directly and their localization is compromised in the absence of PopZ (*Bowman et al., 2010*; *Holmes et al., 2016*). It is therefore conceivable that ZitP also affects motility indirectly from the stalked pole, possibly via cell cycle regulation, flagellar performance and/or polarity. The fact that the motility defect of Δ*zitP* cells can be restored by compensatory mutations in a switch component (FliG) of the flagellar motor (*Kojima and Blair, 2004*), suggests that flagellar performance, reversals or timing (i.e. the length of flagellation in the cell cycle) could be altered by the Δ*zitP* mutation.

Zinc-finger domain proteins other than ZitP may be implicated in linking motility and polarity. The gliding motility protein AgmX confers a flagellum- and pilus-independent form of surface motility in *Myxococcus xanthus* (*Nan et al., 2010*), a δ-proteobacterium that periodically reverses the polarity of movement. Since AgmX also harbors a related N-terminal Zinc-finger domain, at least two related zinc-finger domains control different types of motility. This is intriguing and hints at a potentially important and conserved role of such zinc-finger domain proteins in developmental processes that rely on protein polarization in bacteria and polar matrix proteins such as PopZ to interact with them. In a complementary study, we additionally show *in vitro* and *in vivo* that zinc-bound ZitP binds PopZ directly and regulates PopZ localization without the periplasmic DUF3426 domain (*Bergé et al., 2016*), suggesting that this activity in ZitP may underlie the aforementioned CtrA-independent role in motility.

The conservation of ZitP, CpaM (*Figure 1D*) and PopZ orthologs (*Bowman et al., 2010*) in distant α-proteobacterial lineages that reside in different ecological niches hints that the functions that these proteins control are not unique to the *Caulobacter* branch. Indeed, we describe an interaction between ZitP and PopZ in several distinct α-proteobacterial lineages (*Bergé et al., 2016*). On a more general scale, our work suggests that pole-specific functions conferred by bipolar regulators may be commonly used in bacteria and possibly eukaryotes. Such mechanisms could be relevant for toggle proteins, moonlighting/trigger enzymes (*Commichau and Stulke, 2015*) and other bifunctional regulators (*Radhakrishnan and Viollier, 2012*) that have more than one biochemical activity and function, for example kinase-phosphatases or synthase-hydrolases of cyclic-di-GMP sequestered to both cell poles (*Boyd, 2000*; *Kazmierczak et al., 2006*; *Tsokos and Laub, 2012*).

In sum, the functional and topological versatility of ZitP illustrates how a conserved regulator is used to coordinate multiple functions from different locations and structures in the same cell, relying on distinct protein domains and partners to control localization or to implement function. As these functions and polar remodelling events are coordinated with cell cycle progression in *Caulobacter* via conserved cell cycle proteins, it is likely that superimposed temporal layers similarly act on ZitP and CpaM orthologs in other α-proteobacterial cell cycles.

## Materials and methods

### Strains and growth conditions

*Caulobacter crescentus* NA1000 and derivatives were grown at 30°C in PYE or in M2 salts plus 0.2% glucose (M2G) supplemented with 0.4% liquid PYE (*Ely, 1991*). *Escherichia coli* S17-1, S17-1 λ*pir* and EC100D (Epicentre Technologies, Madison, WI) were cultivated at 37°C in LB. We added 1.5% agar to PYE plates, and motility was assayed on PYE plates containing 0.3% agar. We added D-xylose (0.3% except if otherwise stated), glucose (0.2%), sucrose (3%), kanamycin (solid, 20 µg/ml; liquid, 5 µg/mL), tetracycline (1 µg/mL), spectinomycin (liquid, 25 µg/mL), spectinomycin/

streptomycin (solid, 30 and 5 µg/mL, respectively), apramycin (10 µg/mL), gentamycin (1 µg/mL) and nalidixic acid (20 µg/mL), as required. Swarmer cell isolation, electroporation, biparental mating, and bacteriophage φCr30-mediated generalized transduction were performed as described before (*Chen et al., 2005*; *Ely, 1991*; *Simon et al., 1983*; *Viollier and Shapiro, 2003*).

### Bacterial strains, plasmids, and oligonucleotides

Bacterial strains, plasmids, and oligonucleotides used in this study are listed and described in supplementary tables.

### β-Galactosidase assays

β-Galactosidase assays were performed at 30°C as described previously (*Huitema et al., 2006*; *Viollier and Shapiro, 2003*). Experimental values represent the averages (standard error of the mean, SEM) of at least three independent experiments.

### PALM imaging conditions

To image *C. crescentus*, overnight cultures were diluted in fresh PYE, xylose was added (0.3% final concentration), and the cells were grown for 3 hours to mid-exponential phase (OD (660) ~ 0.4). Two uL of culture was placed on a agarose pad containing PYE. The agarose pad was mounted in a silicone gasket (Grace Biolabs 103280) sandwiched between two microscope coverslips to minimize shrinkage of the agarose. The temperature of the microscope enclosure during experiments was 24°C. Images were acquired using a previously described custom built PALM microscope (*Holden et al., 2014*). Fluorescent proteins were excited at 560 nm, and photoactivation was induced at 405 nm at ~ 0–16 W/cm$^2$. For PALM images of Dendra2-ZitP in *C. crescentus*, cells were imaged at an exposure time of 10 milliseconds for 10,000 frames, and an excitation intensity of ~4 kW/cm$^2$. For stroboscopic single particle tracking PALM measurement of ZitP binding time, cells were imaged at an exposure time of 30 milliseconds, with a variable interval between each frame, at an excitation intensity of ~1 kW/cm$^2$. PALM localizations were accumulated in a 2D histogram; the resulting image was blurred with a 2D Gaussian of radius 15 nm to reflect the localization uncertainty of the measurement. The image was gamma adjusted to 0.5 to compensate for the large dynamic range of the image, and the 'Red Hot' ImageJ colormap was applied.

### Measurement of ZitP binding time by PALM

Binding time, $\tau_{off}$, of ZitP to the *C. crescentus* poles was determined via stroboscopic single particle tracking PALM (*Gebhardt et al., 2013*; *Manley et al., 2008*). Under these conditions, Dendra2 bleached under continuous illumination with a photobleaching lifetime, $\tau_b$, on the order of 50 milliseconds. Since rapid diffusion means that Dendra2-ZitP is only visible when bound to the membrane, and since photobleaching will shorten the observed binding time, the effective on-time of a single Dendra2-ZitP molecule, $\tau_{eff}$, will be the convolution of the photobleaching lifetime, $\tau_b$, and the binding lifetime $\tau_{off}$,

$$\tau_{eff}^{-1} = \tau_{off}^{-1} + \tau_{b}^{-1}, \tag{1}$$

Effective on-time was measured by combining individual Dendra2-ZitP localizations in adjacent frames into tracks (*Crocker and Grier, 1996*), and fitting a single exponential model to the observed the track length distribution (*Figure 3—figure supplement 6A*). In order to measure binding times longer than the photobleaching lifetime, the photobleaching lifetime of the fluorescent protein may be artificially extended by using stroboscopic illumination, introducing large gaps between short periods of illumination. This increases the effective bleaching lifetime to:

$$\tau_{bl}^{'} = \tau_{bl} \frac{\tau_{tl}}{\tau_{int}}, \tag{2}$$

where $\tau_{tl}$ is duration of the gap (time lapse/strobe interval), $\tau_{int}$ is camera integration time. By measuring the effective on-time for multiple different stroboscopic illumination times, $\tau_{tl}$, and performing a fit of:

$$\tau_{eff} = \left(\tau_{\text{off}}^{-1} + \frac{\tau_{int}}{\tau_{bl}\tau_{tl}}\right)^{-1}, \qquad (3)$$

to the data, both the binding time and photobleaching lifetime may be calculated (*Gebhardt et al., 2013*) (*Figure 3—figure supplement 6B and C* Model 1). We performed non-linear least squares fitting of the raw $\tau_{eff}$ data directly to *Eq. 3*, instead of calculating the quantity $\tau_{tl}/\tau_{eff}$ and performing a linear fit as proposed by Gebhardt and coworkers (*Gebhardt et al., 2013*), since the inverse transform proposed results in a non-linear transformation of the sample error distribution incompatible with least squares fitting. We observed that for stroboscopic illumination times significantly greater than the binding time, the data appeared to transition from the hyperbolic relationship predicted by *Eq. 3* to a zero-gradient plateau (*Figure 3—figure supplement 6B*), giving very poor fits between *Eq 3* and the data, especially for the *ΔpopZ* strain which appeared to have a shorter Dendra2-ZitP binding lifetime (*Figure 3—figure supplement 6B*). We hypothesized that this was due to an inability to accurately estimate effective on-time when molecules bind and unbind in a time significantly less than the duration of a single strobing interval (since the observed track length will almost always equal 1 frame). We confirmed this hypothesis by performing the stroboscopic tracking analysis on simulated data (*Figure 3—figure supplement 6C*). We simulated timetraces of molecules binding/unbinding with finite bleaching lifetimes, and measured the observed on-time for each simulated molecule by fitting a single exponential to the on-time histogram as above. We observed as hypothesized that the observed off-times showed a sharp plateau for long-strobe intervals due to the finite integration time of the measurement, giving a poor fit of *Eq 3* to the data (*Figure 3—figure supplement 6C*). In order to correct for this, we modified the fitting model to include a minimum measurable on-time plateau:

$$\tau_{eff} = \left(\tau_{\text{off}}^{-1} + \frac{\tau_{int}}{\tau_{bl}\tau_{tl}}\right)^{-1}, \qquad \tau_{tl} > \tau_{tl}^{min},$$
$$\tau_{eff} = \tau_{tl}^{min}, \qquad otherwise. \qquad (4)$$

Use of the modified model allowed us to obtain accurate fits to the entire simulated dataset (*Figure 3—figure supplement 6C*; *Eq 4*).

We therefore used our updated model to fit the experimental data (*Figure 3E* and *Figure 3—figure supplement 6B*) and to calculate the observed binding times (*Figure 3—figure supplement 6D*). This gave a much better fit to the data, both at late and early strobe intervals. Notably, independent fits to the *WT* and *ΔpopZ* datasets gave similar observed $\tau_{tl}^{min}$ of ~0.4 frames, supporting the use of the updated model.

## Measurement of ZitP cluster area and shape by PALM

In order to estimate the area of Dendra2-ZitP polar complexes, observed localizations were clustered based on local density using DBSCAN (*Endesfelder et al., 2013*; *Ester et al., 1996*). Identified clusters were converted to PALM images binarized, and morphologically closed (*Figure 3—figure supplement 5Bi-iii*). By performing morphological closing on the binary image, we obtained segmented clusters (*Figure 3—figure supplement 5Biii*) which were less sensitive to noise and better reflected the visually estimated extent of the non-segmented cluster. For each identified cluster, the area of the segmented cluster was calculated.

For the NA1000 *xylX::P$_{xyl}$-dendra2-zitP* strain, clusters were visually identified as belonging to the stalked or flagellar poles based on the PALM and phase contrast images of the region. For the *ΔpopZ::Ω xylX::P$_{xyl}$-dendra2-zitP* strain, there was no clear difference in pole morphology, so the cluster area for cells was calculated without discriminating poles. Measurement noise means that the measured area of even a zero-area cluster will be larger than zero (and approximately proportional to the localization uncertainty). To test whether Dendra2-ZitP formed an extended polar complex, we compared the area of ZitP clusters to the measured area of simulated zero-area clusters by generating simulated datasets containing localizations coming from a point source, with photon count, background noise and total number of localizations equal to the median values of either the *WT* or *ΔpopZ::Ω* datasets (*Figure 3—figure supplement 5D*). The cluster area of the simulated datasets was then calculated as above.

We also calculated the following shape based metrics to further quantify the differences in pole shape: *circularity*, *solidity* and *eccentricity* (*Figure 3—figure supplement 5C*).

*Circularity* measures similarity of a shape to a circle, $C = \frac{4\pi A}{p^2}$, where $A$ is shape area and $p$ is perimeter. *Solidity* measures the extent to which a shape is convex or concave, $S = \frac{A}{H}$ where $A$ is shape area and $H$ is the convex hull area of the shape. *Eccentricity* measures how elongated a shape is, $E = \frac{a}{b}$ where $a$ is the length of the minor axis and $b$ is the length of the major axis.

Since the observed distributions showed significant non-normality, statistical significance was assessed by the non-parametric test, Mood's median test. Stars on *Figure 3D* and *Figure 3—figure supplement 5C* indicate statistical significance: n.s, p>0.05; *p<0.05; **p<0.01; ***p<0.001.

The stalked and the other (swamer) pole foci in *WT* showed statistically significant differences (p<*0.001*) in area, circularity and solidity, supporting the conclusion that ZitP forms distinct polar assemblies.

The *WT* stalked pole showed statistically significant differences (p<*0.001*) to the Δ*popZ*::Ω mutant foci for area, circularity, solidity and eccentricity, supporting the conclusion that PopZ specifically promotes the formation of large polar assemblies at the stalked pole.

## Isolation of φCbK resistant mutants

A *himar1*-based transposon mutagenesis of the NA1000 (wild-type, *WT*) strain was done using the *E. coli* S17-1 λ*pir* strain containing the *himar1*-delivery plasmid pHPV414 (*Viollier et al., 2004*). The mutant library was selected on plates containing nalidixic acid and kanamycin embedded in top agar containing φCbK. Colonies emerging from this selection were pooled. We then created generalized transducing lysate from this pool using phage φCr30 and transduced it into strain PV14 Δ*pilA-cpaF*::Ω*aac3* (conferring resistance to aparamycin), selecting for apramycin and kanamycin resistant transductants to eliminate *himar1* insertions in the *pilA-cpaF* locus. The transductants were pooled and a generalized transducing lysate was prepared from this pool using φCr30. This new lysate was then used to transduce NA1000 to kanamycin resistance and the resulting clones were individually tested for resistance to φCbK. The *himar1* insertion site mapping of φCbK–resistant *himar1* mutants was done as described before (*Viollier et al., 2004*).

To isolate the *zitP^{GAP}* mutation, we generated a mutant library of *zitP* alleles by electroporating pMT335-*zitP* into the mutator *E. coli* XL1-Red strain. We collected and pooled over 20,000 clones for plasmid extraction and we electroporated the plasmid library into the Δ*zitP* mutant. We incubated the electroporated cells during two hours for regeneration and next added φCbK for one hour in order to eradicate clones that bear a mutated *zitP* allele restoring effective phage infection. Finally, we plated cells on soft (0.3% swarming) agar to evaluate the motility properties. We picked and streaked out motile clones for amplification and plasmid extraction and introduced the plasmids back into a Δ*zitP* background in the perspective to confirm the motility-proficient and φCbK resistant phenotypes. We isolated a unique plasmid, pMT335-*zitP^{GAP}*, which bears the *zitP^{GAP}* allele (deletion of the Arg93 and Ala94 in the ZitP protein).

## Immunoblotting

Protein samples were separated by SDS-PAGE and blotted on PVDF (polyvinylidenfluoride) membranes (Merck Millipore). Membranes were blocked for 1 hour with Tris-buffered saline, 0.05% Tween 20 (TBST), and 5% dry milk and then incubated for an additional 1 hour with the primary antibodies diluted in TBST, 5% dry milk. The membranes were washed 4 times for 5 minutes in TBST and incubated for 1 hour with the secondary antibody diluted in TBST, and 5% dry milk. The membranes were finally washed again 4 times for 5 minutes in TBST and revealed with Immobilon Western Blotting Chemoluminescence HRP substrate (Merck Millipore) and Super RX-film (Fujifilm). Rabbit antisera were used at the following dilutions: anti-CtrA (1:10,000), anti-PilA (1:10,000), anti-FljK (1:50,000), anti-CpaC (1:5000), anti-ZitP (1:5000), anti-CpaM (1:5000) and anti-GcrA (1:2000). HRP-conjugated Anti-rabbit secondary antibody was used at 1:20,000 dilution (Jackson ImmunoResearch, USA).

## Epi-fluorescence microscopy

PYE or M2G cultivated cells in exponential growth phase were immobilized using a thin layer of 1% agarose. Fluorescence and DIC images were taken with an Alpha Plan-Apochromatic 100x/1.46 DIC

(UV) VIS-IR oil objective on an Axio Imager M2 microscope (Zeiss) with 488 nm laser (Visitron Systems GmbH, Puchheim, Germany) and a CoolSnap HQ (*Boutte et al., 2012*) camera (Photometrics) controlled through Metamorph V7.5 (Universal Imaging). Images were processed using Image J software. Quantifications were done by manually numbering cells in the diffuse, monopolar or bipolar state.

## Protein purification and production of antibodies

The PCR-amplified *zitP^Cterm* and *cpaM^ΔTM* genes were cloned into the pET28a vector (Novagen). The His$_6$-ZitP^Cterm and His$_6$-CpaM^ΔTM recombinant proteins were overexpressed *in E. coli* strain Rosetta and purified in standard native conditions on Ni$^{2+}$-NTA agarose as described previously to raise rabbit polyclonal IgGs in New Zealand White rabbits (Josman LLC, Napa, CA).

## Tandem affinity purification (TAP) and mass spectrometry

We followed the TAP procedure as was previously described (*Puig et al., 2001*). When a 1 L-culture reached OD660 between 0.4 and 0.6 in the presence of 50 mM vanillate, cells were harvested by centrifugation (6000xg, 10 min). We washed the pellet in 50 mL of buffer I (50mM sodium phosphate pH 7.4, 50 mM NaCl, 1 mM EDTA) and lysed for 15 minutes at room temperature in 10 mL of buffer II (buffer I + 0.5% n-dodecyl-β-D-maltoside, 10mM MgCl$_2$, two protease inhibitor tablets [Complete EDTA-free, Roche] per 50 mL of buffer II, 1x Ready-Lyse lysozyme [Epicentre], 500U of DNase I [Roche]). Cellular debris was removed by centrifugation (7000xg, 20 minutes, 4°C). The supernatant was incubated for 2 hours at 4°C with IgG Sepharose beads (GE Healthcare Biosciences) that had been washed once with IPP150 buffer (10 mM Tris-HCl pH 8, 150 mM NaCl, 0.1% NP40). After incubation, the beads were washed at 4°C three times with 10 mL of IPP150 buffer and once with 10 mL of TEV cleavage buffer (10 mM Tris-HCl pH 8, 150 mM NaCl, 0.1% NP40, 0.5 mM EDTA, 1 mM DTT). The beads were then incubated overnight at 4°C with 1 mL of TEV solution (TEV cleavage buffer with 100 U of TEV protease per ml [Promega]) to release the tagged complex. 3 mM CaCl$_2$ was then added to the solution. The sample with 3 mL of calmodulin-binding buffer (10 mM β-mercaptoethanol, 10 mM Tris-HCl pH 8, 150 mM NaCl, 1 mM magnesium acetate, 1 mM imidazole, 2 mM CaCl$_2$, 0.1% NP40) was incubated for 1 hour at 4°C with calmodulin beads (GE Healthcare Biosciences) that previously had been washed once with calmodulin-binding buffer. After incubation, the beads were washed three times with 10 mL of calmodulin-binding buffer and eluted five times with 200 µL IPP150 calmodulin elution buffer (calmodulin-binding buffer substituted with 2 mM EGTA instead of CaCl$_2$). Amicon Ultra-4 spin columns (Ambion) were used to concentrate eluates. Eluates were analyzed by SDS-PAGE and stained with silver using the Biorad Silver Stain Plus kit (Biorad, USA). We then cut specific bands and directly sent the gel slices to the Taplin Biological Mass Spectrometry Facility (Harvard Medical School, Boston, USA) for mass spectrometric analyses.

## Co-immunoprecipitation

Cells were harvested from a 50 mL-culture (OD (660 nm) between 0.4–0.6) by centrifugation at 5000xg for 10 minutes. We washed the cell pellet in 10 mL of buffer I (50mM Tris-HCl (pH 7.5); 50 mM NaCl; 1mM EDTA), centrifuged the cell again and resuspended in 1 mL of buffer II (buffer I plus 0.5% n-dodecy-β-D-maltoside; 10 mM MgCl$_2$; EDTA-free protease inhibitors). We incubated the mixture for 15 minutes with 5000 units of Ready-Lyse lysozyme (Epicentre), and 30 units of DNase I (Roche). Cellular debris were removed by centrifugation at 10,000xg for 3 minutes at 4°C. We cleared the supernatant by incubation for 1 hour at 4°C with Protein-A agarose beads (Roche) previously washed three times with 500 µL of buffer II. We removed agarose beads by centrifugation and added to the pre-cleared solution polyclonal IgG rabbit serum for 90 min at 4°C (dilution 1:500). Next, we trapped for 1 hour at 4°C the antibodies-proteins complexes with the addition of Protein-A agarose beads (Roche) previously washed three times with 500 µL of buffer II. The samples were then centrifuged at 3000xg for 1 minute at 4°C and the supernatant was removed. The beads were washed once with buffer I plus 0.5% n-dodecy-β-D-maltoside, three times with 500 µL of wash buffer (10 mM Tris-HCl at pH 7.5; 150 mM NaCl; 0.1% n-dodecy-β-D-maltoside) and finally resuspended in 70 µl SDS sample buffer (50 mM Tris–HCl at pH 6.8), 2% SDS, 10% glycerol, 1% β-mercaptoethanol, 12.5 mM EDTA, 0.02% Bromophenol Blue), heated to 95°C for 10 minutes and stored at −20°C.

## Motility assays and phage infectivity tests

Swarming properties were assessed with 5 µl-drops of overnight culture spotted on PYE soft agar plates (0.3% agar) and grown for 24 hours. Phage susceptibility assays were conducted by mixing 500 µL of overnight culture in 6 mL soft PYE agar and overlaid on a PYE agar plate. Upon solidification of the soft (top) agar, we spotted 5 µL-drops of serial dilution of phages (φCbK or φCr30) and scored for plaques after one day incubation at 4°C.

## Shearing experiments

We centrifuged 5 mL mid-log phase cultures of *WT* or mutant strains and resuspended them in 700 µl of PYE. Then, we pumped in and out (10x) the cells into a syringe endowed with a thin diameter needle. We centrifuged the shear-stressed cells to remove cells debris and collected 200 µL of each supernatant. We added SDS-blue straining and loaded samples on SDS-PAGE gels.

## $\varphi$CbK adsorption assay

To determine the adsorption rate of φCbK, *Caulobacter crescentus* NA1000 and derivatives were first grown overnight in M2G medium at 30°C and then re-started in fresh M2G at 30°C with shaking until the bacterial cell culture reached an OD660 value of 0.4 ($0.4 \times 10^8$ CFU/ml). Then cell cultures were infected by 0.02 multiplicity of φCbK infection (MOI: ratio of phage to bacteria number). The mixtures were incubated at 30°C without shaking for phage adsorption, followed by separation of unbound phages by centrifugation at 13,000 rpm in specified time points up to 30 minutes. Supernatants were immediately supplemented by the addition of chloroform (1/20 of cell culture volume) and mixed vigorously to kill remaining bacterial cells. A control tube containing only φCbK (equivalent to 0.02 MOI) was maintained in parallel for the duration of the experiment and used as reference to control the initial phage plaque-forming units (pfu) titer. A 50 µL of the phage supernatant from each tube was mixed with 200 µL of *Caulobacter crescentus* NA1000 culture at log phase and incubated without shaking at room temperature for 10 minutes to allow adsorption. Infected cells were added to 6 mL of soft PYE agar (0.5%) and immediately overlaid on 1.5% PYE agar plates. Plates were incubated at 30°C for 24 hours, when pfu were visible. The φCbK adsorption value (in% of the initial phage pfu titer) was calculated. Values are the mean of three biological replicates; error bars represent data ranges.

## Flow cytometry (Fluorescence-activated cell sorting, FACS)

Cells in exponential growth phase (OD660nm=0.3–0.6) cultivated in PYE, were fixed in ice-cold 77% ethanol solution. Fixed cells were re-suspended in FACS staining buffer, pH 7.2 (10 mM Tris-HCl, 1 mM EDTA, 50 mM NaCitrate, 0.01% Triton X-100) and then treated with RNase A (Roche) at 0.1 mg mL$^{-1}$ for 30 minutes at room temperature. Cells were stained in FACS staining buffer containing 0.5 µM of SYTOX Green nucleic acid stain solution (Invitrogen) and then analysed using a BD Accuri C6 flow cytometer instrument (BD Biosciences). Flow cytometry data were acquired and analysed using the CFlow Plus V1.0.264.15 software (Accuri Cytometers Inc.). 20,000 cells were analysed from each biological sample. Experimental values represent the averages of three independent experiments.

## *fliG*$^{D306G}$ swarming pseudo-revertant isolation and backcrossing

We spotted several 5 µL-drops of Δ*zitP* overnight culture on soft agar plates and waited for flares spreading out the bulk of cells. Flares were peaked out and streaked on fresh agar plates for amplification and subsequently challenged for motility in comparison to *WT* and Δ*zitP* strains. Motility-proficient clones were sent for Illumina HiSeQ 2000 sequencing (Fasteris, www.fasteris.com/). Genomes were compared to NA1000 genome and we identified a single mutation in the *fliG* gene (D306G).

In order to backcross the *fliG*$^{D306G}$ allele in different backgrounds, the suppressor strain was electrotransformed with the suicide vector pNTPS138-hook and selected on kanamycin-supplemented plates for single crossing-over in close vicinity of the *fliG* locus. We prepared lysate of this strain, transduced the *fliG*$^{D306G}$-linked pNTPS138 into *WT* and Δ*zitP* cells and screen by sequencing for clones harbouring the *fliG*$^{D306G}$ allele. Finally, we grew up the strain without any antibiotic and selected for plasmid excision by plating an overnight culture on sucrose.

## Acknowledgements

Funding support was from Swiss National Science Foundation grants to PHV, European Research Council Starting Grant 243016 to SM and Marie Curie Intra-European Fellowship PIEF-GA-2011-297918 to SH. We are grateful to Florian Gays for technical support with phage adsorption assays.

## Additional information

### Funding

| Funder | Grant reference number | Author |
|---|---|---|
| Schweizerischer Nationalfonds zur Förderung der Wissenschaftlichen Forschung | 31003A_162716 | Patrick H Viollier |
| European Research Council | 243016 | Suliana Manley |
| Marie Curie Intra-European Fellowship | PIEF-GA-2011-297918 | Seamus Holden |

The funders had no role in study design, data collection and interpretation, or the decision to submit the work for publication.

### Author contributions

JM, SH, MB, SM, Conception and design, Acquisition of data, Analysis and interpretation of data, Drafting or revising the article, Contributed unpublished essential data or reagents; GP, Conception and design, Acquisition of data, Analysis and interpretation of data, Contributed unpublished essential data or reagents; EE, Conception and design, Acquisition of data, Analysis and interpretation of data; LT, Acquisition of data, Contributed unpublished essential data or reagents; PHV, Drafting of the work, Acquisition, Analysis, Interpretation of data for the work, Conception and design, Acquisition of data, Analysis and interpretation of data, Drafting or revising the article, Contributed unpublished essential data or reagents

### Author ORCIDs

Johann Mignolet, http://orcid.org/0000-0002-3721-4307
Seamus Holden, http://orcid.org/0000-0002-7169-907X
Patrick H Viollier, http://orcid.org/0000-0002-5249-9910

## Additional files

### Supplementary files

• Supplementary file 1. Bacterial strains, plasmids and oligonucleotides list

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
