## [Decision Letter]

Thank you for submitting your article "Functional dichotomy, distinct nanoscale assemblies and cell cycle control of a bipolar zinc-finger regulator" for consideration by *eLife*. Your article has been reviewed by three peer reviewers, and the evaluation has been overseen by a Reviewing Editor and Vivek Malhotra as the Senior Editor. One expert involved in review of your submission, Jan Willem Veening (Reviewer #1), has agreed to reveal his identity.

The reviewers have discussed the reviews with one another and the Reviewing Editor has drafted this decision to help you prepare a revised submission.

Mignolet and co-workers show, using a combination of genetics, biochemistry and super resolution microscopy techniques, how a protein (ZitP) can perform two distinct biological functions at each end of the cell pole. They discover a distinct assembly of ZitP, only visible by nanoscopy, at each end of the *Caulobacter* pole, the stalked end and the piliated end. Compelling evidence is provided that this distinct assembly of ZitP recognizes specific other proteins and in that way control pilus biogenesis and swarming motility. Finally, it is shown that ZitP is hard-wired in the cell cycle and is not produced in the G1 to S phase. All reviewers found interest in the study but they compiled a list of necessary improvements before the paper can be published.

Essential revisions:

1) In this version, the paper is extremely compact making it difficult to follow the logical flow for the non-*Caulobacter* expert. The take-home message itself is difficult to extract due to the lack of background and numerous different messages. We believe that the important finding here is the description of a novel class of potentially widespread polarity regulators.

To improve clarity, the authors should use more space, separate figures and reduce the emphasis on the cell cycle itself (one figure could be specifically dedicated to the cycle) and add more knowledge about polarity control in bacteria. The predicted structure of ZitP itself is quickly glossed over. Why was it even called ZitP? The authors mention the presence of a Zinc atom in ZitP but what is the evidence supporting this? Please provide a complete bioinformatics view of ZitP and homologs in other procaryotes (and beyond?). The other key players should also be described for understanding by a generalist audience, the pilus assembly pathway, CpaC, CpaC*, PopZ etc…

2) The PALM analysis is not used to its optimum to show that ZitP is part of two distinct nanoscale assemblies. Further quantitative analysis should improve the description of the clusters and the difference in popZ mutant cells. How many molecules, are the shapes significantly different (the stats indicate so but the graphs are not that convincing), can they be further resolved? The single particle PALM analysis needs to be better explained, it is not clear whether the binding times are similar at both poles in WT and why the longer dwelling times were ignored in the fit shown in the Figure 2—figure supplement 2?

3) The genetic data showing that function of ZitP can be uncoupled is clear and interesting. How Zitp affects pilus assembly is well documented with the identification of an interaction with CpaM (but how CpaM fits in the assembly pathway could be clarified by better presentation of the assembly pathway). On the other hand, how Zitp affects swarming and how this is linked with differential localization is not clear, especially because the *Caulobacter* flagellum does not localize to the stalk pole. How is swimming affected in the ZitP swarm- mutant? The authors suggest that the swarming defect could be due to the lower number of G1 cells. Is ZitP involved in cell cycle regulation, how?

4) In the final cartoon, again what is the evidence that ZitP binds Zinc and that it adopts open and closed conformation? Why is the closed conformation shown to bind popZ at both poles? Is there experimental data to support this proposal? What is the localization of the ZitP-GAP mutant? Overall the ZitP mutants and the expected changes linked to function need to be better described.

---

## [Author Response]

[…]

*Essential revisions:*

*1) In this version, the paper is extremely compact making it difficult to follow the logical flow for the non-Caulobacter expert. The take-home message itself is difficult to extract due to the lack of background and numerous different messages. We believe that the important finding here is the description of a novel class of potentially widespread polarity regulators.*

*To improve clarity, the authors should use more space, separate figures and reduce the emphasis on the cell cycle itself (one figure could be specifically dedicated to the cycle) and add more knowledge about polarity control in bacteria. The predicted structure of ZitP itself is quickly glossed over. Why was it even called ZitP? The authors mention the presence of a Zinc atom in ZitP but what is the evidence supporting this? Please provide a complete bioinformatics view of ZitP and homologs in other procaryotes (and beyond?). The other key players should also be described for understanding by a generalist audience, the pilus assembly pathway, CpaC, CpaC*, PopZ etc.*

We are grateful to the reviewing editor as well as the reviewers for the constructive recommendations on how to improve clarity and comprehensiveness in our manuscript. The enclosed version is now substantially reorganized and augmented with new experiments that reveal new insight on the action of ZitP controlling CtrA-activated (cell cycle regulated) promoters via CpaM. Moreover, we used genetic pseudoreversion analysis to correct this defect in CtrA activity and to reveal another role of ZitP on motility, clearly establishing its multifunctional nature (Figure 2).

We also detail a new mutation, ZitP^1-133^, that has the opposite effect than the previously described ZitP(CS) mutation in that it affects pilus biogenesis and CtrA activity via CpaM, but not motility. We restructured figures according to the text flow and by theme: Figure 1 – bioinformatics and primary structures, Figure 2 – function, Figure 3 – localization and Figure 4 – cell cycle control. Finally, we bolstered Figure 1 with a phylogenetic tree to show ZitP, as well as CpaM, conservation across α-proteobacteria (Figure 1).

Indeed, this manuscript does *not* provide direct evidence that ZitP (zinc-finger targeting the poles) coordinates a zinc ion. However, in complementary study (Bergé et al., submitted to *eLife,* 2016) we determined the structure –function analyses and resolved the tertiary structure of the zinc-coordination module by liquid state NMR. Thus, ZitP does indeed bind a zinc ion through the four conserved cysteine residues. The revised version that we submit here now refers extensively to this complementary data in the Discussion.

*2) The PALM analysis is not used to its optimum to show that ZitP is part of two distinct nanoscale assemblies. Further quantitative analysis should improve the description of the clusters and the difference in popZ mutant cells. How many molecules, are the shapes significantly different (the stats indicate so but the graphs are not that convincing), can they be further resolved?*

In response to the comment on the graphs not being convincing, our understanding of this comment is that a clear difference is not visible between the poles in the images of ZitP localization presented in the old Figure 2. We additionally 1) provide multiple zoomed in representative example images of polar foci, showing the visible differences in shape and size (new Figure 3 and Figure 3—figure supplement 2, 2C and 5C). The zoomed in figures show clearly visible differences in ZitP cluster shape between WT stalked pole, WT swarmer pole, and the poles of △*popZ* cells.

In new Figure 3, the cluster area plot shows a visible and statistically significant difference in ZitP cluster area between the swarmer and stalked poles as supported by the highly significant difference observed (p < 10^-8^). For additional clarity, we have re-plotted the data as a violin plot and moved the “both WT poles” comparison and the “zero-area” comparison to Figure 3—figure supplement 5. We note that the statistical test we applied is a very conservative non-parametric test (Mood’s median test), so such a high statistical significance is particularly strong evidence that the difference is genuine.

We performed additional quantitative analysis of the PALM data to confirm the observed difference in ZitP swarmer/stalk polar assemblies: we analysed the cluster shape (circularity, solidity, eccentricity) in addition to the current measurement of area (Figure 3—figure supplement 5), and additional discussion (in Materials and methods). The additional quantitative shape-based analysis confirms with high significance the previously observed difference between ZitP assemblies at stalked and swarmer poles in WT.

We did not perform molecular counting or stoichiometry analysis on these data because Dendra2-ZitP was expressed from an inducible plasmid, meaning that molecule counts at each pole will not be an accurate reporter of absolute wild type levels of ZitP, and stoichiometry analysis will be unreliable due to cell-cell variation as cells are alive and constantly turn-over molecules during data acquisition. Since such data would be unreliable and difficult to interpret, we focussed on shape-based cluster analysis described above.

*The single particle PALM analysis needs to be better explained, it is not clear whether the binding times are similar at both poles in WT and why the longer dwelling times were ignored in the fit shown in the Figure 2—figure supplement 2?*

The reviewers highlighted an important issue with the binding time analysis – using the previously published approach it was necessary to ignore longer strobe intervals in order to get reasonable fits to the data. In the previous submitted manuscript version we hypothesized that this was a measurement artefact – essentially, at long time-lapse intervals, molecules will almost always have unbound within a single frame of observation (‘saturation’). In this case it becomes impossible to accurately measure unbinding time, hence the artefactual flat line trend. However, we did not prove this hypothesis, and we agree that the manual exclusion of late datapoints is very unsatisfying. We thank the reviewer for their recommendation to pursue this issue further.

We first confirmed by simulation that the cause of the long-time-lapse artefact was indeed saturation (Materials and methods, Figure 3—figure supplement 6). We completely revised and updated the analysis method SPT PALM to account for this issue (Materials and methods, Figure 3—figure supplement 6), and with our updated method were able to obtain robust fits to the entire dataset (Figure 3 and Figure 3—figure supplement 6). This not only improves our analysis of ZitP binding time, but significantly improves the robustness of the stroboscopic SPT PALM method as a general tool.

At the reviewer’s suggestion, we attempted to perform analysis of the binding time for the WT swarmer and WT stalked poles separately. However, it was not possible to identify sufficient clearly distinguishable stalked or swarmer poles in the collected data to perform a statistically robust analysis. This is because insufficient localizations of individual molecules could be collected for the binding time analysis on a per-pole basis due to the long frame intervals (seconds) required, and thus the difficulty of knowing which pole is which when examining a few single molecules for a long period. By contrast for the shape-based analysis, many localizations for separate poles could be acquired due to the high frame rate (15ms).

Although the per-pole binding time analysis was not possible, the visually observed morphological differences in WT swarmer/ WT stalk/ δ-PopZ polar assemblies (Figure 3, Figure 3—figure supplement 2), supported by quantitative shape based analysis (Figure 3—figure supplement 5), together with the biological evidences (different intrinsic and extrinsic polar determinants) already provide strong evidence for distinct ZitP polar assemblies.

*3) The genetic data showing that function of ZitP can be uncoupled is clear and interesting. How Zitp affects pilus assembly is well documented with the identification of an interaction with CpaM (but how CpaM fits in the assembly pathway could be clarified by better presentation of the assembly pathway). On the other hand, how Zitp affects swarming and how this is linked with differential localization is not clear, especially because the Caulobacter flagellum does not localize to the stalk pole. How is swimming affected in the ZitP swarm- mutant? The authors suggest that the swarming defect could be due to the lower number of G1 cells. Is ZitP involved in cell cycle regulation, how?*

As requested, we elaborated on what is known about the *Caulobacter* pilus assembly (Cpa) pathway in the Introduction and we included a corresponding scheme (Figure 1) to summarize the specific role of ZitP/CpaM. We also significantly enriched the manuscript with data and discussion about the multifactorial (indirect) effects of ZitP on motility. In *Caulobacter*, motility is a multi-step and coordinated process that relies on direct (flagellum biogenesis, chemotaxis) and indirect (cytokinesis, DNA replication/segregation, polarity and cell cycle) events. We now show that the motility can be restored by compensatory mutations in the flagellar motor, suggesting that flagellar performance, reversal or length of flagellation in the cell cycle is altered. Indeed we observed a decrease in CtrA- (but not GcrA-)dependent promoters in △*zitP* cells versus WT(Figure 2—figure supplement 1), a defect that is also seen in △*cpaM* cells. However, as △*cpaM* cells only exhibit a mild motility defect compared to △*zitP* cells (Figure 2), the reduced CtrA-dependent transcription only contributes in minor fashion to motility. Indeed, when the CtrA-defect in △*zitP* is corrected by ectopic expression of (p)ppGpp, motility is mildly improved but still clearly discernible (Figure 2—figure supplement 2). We reason that there is another unknown role of ZitP to motility and that this could involve by the role of ZitP in regulating the localization of the polar matrix protein PopZ as described in the companion paper (Bergé et al., submitted to *eLife,* 2016). Alternatively, ZitP may regulate another pathway, directly or indirectly, that remains to be identified.

*4) In the final cartoon, again what is the evidence that ZitP binds Zinc and that it adopts open and closed conformation? Why is the closed conformation shown to bind popZ at both poles? Is there experimental data to support this proposal? What is the localization of the ZitP-GAP mutant? Overall the ZitP mutants and the expected changes linked to function need to be better described.*

In (Bergé et al., submitted to *eLife,* 2016) we showed that ZitP coordinates a zinc ion via the four cysteines of the zinc-finger domain. ZitP^CS^ does not localize at the stalked pole, while PopZ recruits ZitP at the same location, meaning that zinc binding and PopZ are the intrinsic and extrinsic determinants for ZitP stalked pole-accumulation, respectively. We localized ZitP (or CpaM) in all polar factor mutants inventoried in *Caulobacter* (unpublished data) but we did not find the extrinsic determinant that targets ZitP (and so CpaM) to the swarmer pole. Hence, we cannot rule out the possibility that PopZ also recruits ZitP at the swarmer pole in native conditions and we revised our model, including a question mark close to ZitP(Zn2+) at the swarmer pole (Figure 4). In wild-type cells, it would be unlikely that the ZitP(no Zn2+) form is recruited at the stalked pole. Indeed, in such a case, ZitP^CS^ and CpaM should be present at the stalked pole as well.

We included experimental data concerning the ZitP^GAP^ localization. Considering that this variant still interacts with and recruits CpaM at the swarmer pole, ZitP^GAP^ is consistently still located at both poles. However, ZitP^GAP^ does not restore phage sensitivity, suggesting that CpaM localization is necessary but not sufficient for pilus biogenesis and that ZitP has an extra role, possibly through CpaM activation.